# Relationship of Textures from Tomato Fruit Images Acquired Using a Digital Camera and Lycopene Content Determined by High-Performance Liquid Chromatography

Ewa Ropelewska * and Justyna Szwejda-Grzybowska

Fruit and Vegetable Storage and Processing Department, The National Institute of Horticultural Research, Konstytucji 3 Maja 1/3, 96-100 Skierniewice, Poland
* Correspondence: ewa.ropelewska@inhort.pl

**Abstract:** This study aimed at correlating image features with the lycopene content of tomato fruit. Tomato cultivars with different fruit colors, such as 'Ożarowski' (yellow), 'Marvel Striped' (yellow-orange-pink), 'Green Zebra' (green), Sandoline F1 (red), Cupidissimo F1 (red), and Sacher F1 (brown) were selected for the study. The tomato fruits were imaged using a digital camera. The texture parameters were computed from the images converted to color channels *R*, *G*, *B*, *L*, *a*, *b*, *X*, *Y*, and *Z* based on the histogram, autoregressive model, gradient map, co-occurrence matrix, and run-length matrix. Lycopene content was determined using high-performance liquid chromatography (HPLC). Pearson's correlation coefficients (R), regression equations, and coefficients of determination ($R^2$) were determined. The lycopene content in fruit ranged from 0.31 mg 100 g$^{-1}$ for 'Green Zebra' to 11.83 mg 100 g$^{-1}$ for Sacher F1. The correlation coefficient (R) between lycopene content and selected image textures reached $-0.99$ for selected textures from color channels *G*, *b*, and *Y*. The highest positive correlation (R parameter equal to 0.98) was obtained for texture from color channel *Y*. Based on the individual color channel providing the highest results, one texture was selected for the determination of regression equations. Coefficients of determination ($R^2$) of 0.99 were obtained for texture from color channel *G*. The regression equations may be used in practice for nondestructive, objective, and precise estimation of the lycopene content in tomato fruit.

**Keywords:** tomato cultivars; fruit color; tomato lycopene; image analysis; correlation; regression

## 1. Introduction

Tomato (*Solanum lycopersicum* L.) is a species of plant in the nightshade family. It comes from Central America and came to Europe in the 16th century. Initially, it was grown only for ornamentation, as it was considered a poisonous plant. Currently, it is highly appreciated by consumers and is valued not only for its taste and culinary qualities, but mainly for its health and dietary properties and low energy value [1]. Tomato fruits are an important source of pro-health compounds such as carotenoids (mainly lycopene, β-carotene, α-carotene, lutein, zeaxanthin), phenolic compounds, and vitamins such as ascorbic acid, niacin, biotin, thiamin, riboflavin, pantothenic acid and folate, and vitamin K1 [2,3]. These cover 50 to 120% of the recommended daily intake of vitamin C, 10 to 30% of vitamin A and 12% of vitamin E [3].

Currently, there are over 15,000 varieties of tomatoes in the world, mainly due to the growing market demand. In recent times, there has been an increase in the awareness of consumers and the medical world about healthy eating. Today, attention is paid to what constitutes the composition of food, because in addition to the basic nutritional value, what we eat can have a positive effect on health. The substances contained in fresh tomato fruit and their preserves play an important role in the prevention of cardiovascular diseases and cancer [4–6]. Among these substances, the most important is lycopene, which has strong health-promoting properties, and participates in the scavenging of singlet oxygen and

peroxide radicals [7]. It is also involved in the regulation of the cell cycle and the induction of programmed cell death. This compound is increasingly more often treated not only as a dietary supplement but also as a potential medicine. It also affects the hormonal balance and the body's immunity [8]. Lycopene accounts for about 80–90% of the pigments found in tomato fruits, the remainder being β-carotene and other carotenoids. It is located mainly in the peel of tomatoes, where it is about five times more than in the flesh. It accumulates mainly during the final period of fruit ripening [9,10]. The carotenoid content can be determined, for example, using spectrophotometric measurements, but high-performance liquid chromatography (HPLC) can be characterized by its higher accuracy for identifying and quantifying individual carotenoids from the extracts [11]. The lycopene content in tomatoes is commonly determined by HPLC [12,13]. Numerous studies show that lycopene is better absorbed from processed products than from fresh raw materials. The use of high temperature for technological processing and for preserving ready-made products destroys the cell walls of fresh tomato fruits. This causes lycopene to be easily released from the cell juice [14–16]. The largest number of fresh tomatoes in the world is consumed in North America—about 40 kg per person. The second region with high consumption of tomatoes is Asian countries. In the case of European countries, consumption is approximately 30 kg per person per year [17].

The tomato cultivars can differ in color, size, shape, and fruit flavor. The color and shape of the fruit in the phase of consumption maturity depend on the presence of a system of genes—biological information stored in the cells of every living organism. The color of tomato buds is always green, and other colors appear only as the fruit ripens. Chlorophyll disappears, and as a result of natural transformations, the synthesis of carotenoids occurs: lycopene and ζ-carotene (red pigment), β-carotene (orange pigment), γ-carotene (pink pigment), α-carotene, lutein or zeaxanthin (yellow pigment). Sometimes during maturation, purple anthocyanins appear in the form of streaks or complete dark purple discoloration of the skin. The disappearance of chlorophyll may be so slow that the fruit becomes consumable before the carotenoids appear, or it may be so low that the fruit appears white. The most common red color in tomato fruits is due to the gene responsible for the synthesis of lycopene. Other genes are responsible for the yellow color or suppression of red carotenoids. Raspberry tomatoes have pink γ-carotene and few yellow pigments. Two genes limiting the formation of lycopene in the fruit are responsible for the orange color. Yellow and orange pigments become visible only during the fruit's over ripening, and the genes responsible for them completely dominate the "red" gene. Various natural genetic variations, sometimes due to exceptional stress conditions (e.g., viruses), are responsible for uneven maturation. This is generally a disadvantage but if the fruit looks attractive, growers consolidate these features through appropriate crosses. Then, the tomatoes are shaded, striped red-yellow, green-green, green-brown, or golden and white spots. Different cultivars may have the same fruit color but vary in other characteristics. It happens, however, that the color is the only difference—then a cultivar is created in several color versions [18–20]. Considering color, image texture, and geometric parameters, the cultivar discrimination of fruit can be performed using nondestructive, objective, and fast procedures involving image processing and machine learning [21–23].

Owing to their size, tomatoes are divided into small-fruit cultivars with fruit up to 50 g (cocktail, cherry, bead), mid-fruit cultivars—most cultivars up to 170 g, and large-fruit cultivars (raspberry, buffalo heart, gargamel) with fruit weight of 200–300 g. The taste and smell, structure of the flesh and skin, and juiciness and durability after harvest are determined by both genetic characteristics and growing conditions. The shape of the tomatoes is an equally important varietal feature. The basic shapes of fruit in cultivars are ball, oval and flattened, or ribbed (inside chambers are marked). In the process of crossing and selection, unforeseen features become apparent over time and fruit with unusual shapes is obtained. Thus, many cultivars with different degrees of ball flattening or elongation were obtained: plum-shaped, long and narrowed (San Marzano type) or resembling a pointed pepper, pear-shaped, heart-shaped, more or less ribbed, including

pouch-shaped, and even quite irregular. The shape does not affect consumption values. Both genetic characteristics and growing conditions determine the taste and smell, flesh and skin structure, juiciness, and shelf life [24].

The objective of this study was to correlate image features of tomato fruit belonging to different cultivars with the lycopene content. By determining the relationship between the features of tomato images and the lycopene content, it is possible to estimate the content of lycopene without the need for destructive, more expensive, and time-consuming measurements. The regression equations developed may allow for precise, objective, and nondestructive determination of the lycopene content in tomato fruit, which is the great novelty of this research and may be of practical use in food processing and tomato consumption.

## 2. Materials and Methods

### 2.1. Material

The research material was six cultivars of tomato: 'Ożarowski' (yellow), 'Marvel Striped' (yellow-orange-pink), 'Green Zebra' (green), Sandoline F1 (red), Cupidissimo F1 (red), and Sacher F1 (brown). The tomatoes from potted seedlings were grown in an unheated tunnel from April to September. All tomato fruit cultivars were purchased from the producer in the last week of June, in the consumption maturity phase. For each cultivar, fifty fruits were obtained. For the analysis of lycopene, samples were finely sliced and frozen prior to freeze-drying, after which they were powdered and stored at $-20$ °C.

### 2.2. Image Analysis

The imaging system used in the experiment consisted of a digital camera mounted on a tripod and placed in a black box with LED (light emitting diodes) illumination. The digital camera included optical image stabilization, auto white balance, F 2.4, $8\times$ digital zoom, and SD (secure digital) card. The LED illumination was characterized by the light sources of 24 LED, related input voltage of AC 110–240 V at 50–60 Hz, related input current of 0.07 A, and related output power of 2.2 W [25]. The tomato fruits were imaged against a black background. Each tomato was imaged individually, and each image was saved separately. Fifty images for each of the 'Ożarowski', 'Marvel Striped', 'Green Zebra', Sandoline F1, Cupidissimo F1, and Sacher F1 tomato cultivars were acquired. The sample tomato images are presented in Figure 1.

The images were uploaded to a computer using a USB cable. The image processing was performed using the Mazda software (Łódź University of Technology, Institute of Electronics, Łódź, Poland) [26]. First, the images were converted to individual color channels $R$, $G$, $B$, $L$, $a$, $b$, $X$, $Y$, and $Z$. The tomato images from selected color channels are shown in Figure 2.

The image segmentation was carried out using the thresholding method to separate each object from the black background and overlay the ROI (region of interest). About 200 textures based on the histogram, autoregressive model, gradient map, co-occurrence matrix, and run-length matrix were determined for each ROI [26]. Textures determined in this study are considered as numerical data extracted from images and are defined as a function of the spatial variation of the brightness intensity of the pixels. Objects can be characterized by different textures even if they have the same color histograms and number of pixels but a dissimilar color distribution. These changes can be difficult to perceive visually [27,28]. After imaging, the tomatoes were used to determine the lycopene content. The flowchart for the stages involved in the proposed procedure for determining the relationship between image textures of tomato fruit and lycopene content is presented in Figure 3.

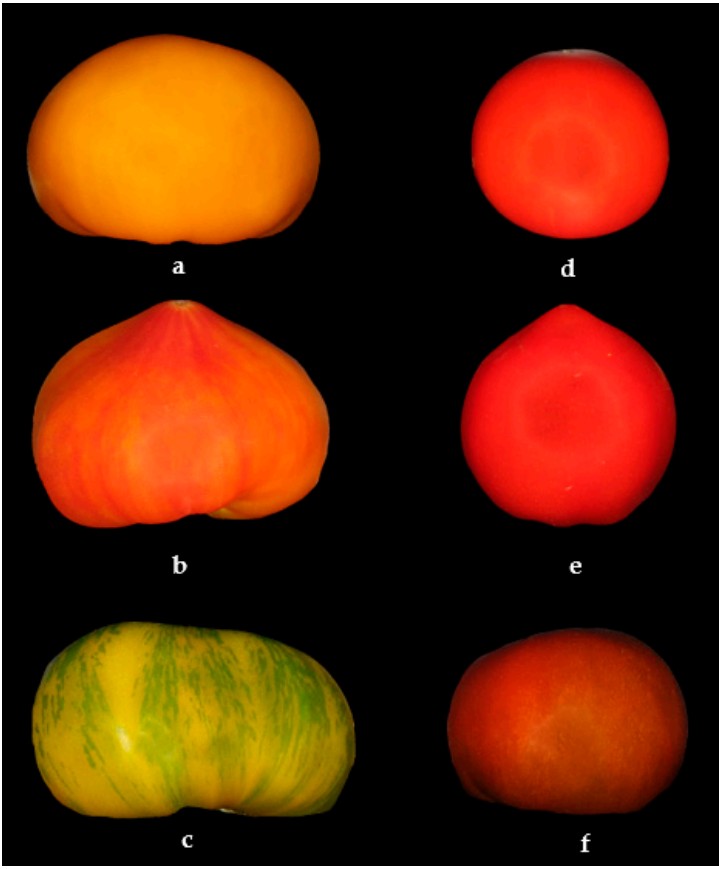

**Figure 1.** The sample color images of tomatoes 'Ożarowski' (**a**), 'Marvel Striped' (**b**), 'Green Zebra' (**c**), Sandoline F1 (**d**), Cupidissimo F1 (**e**), and Sacher F1 (**f**).

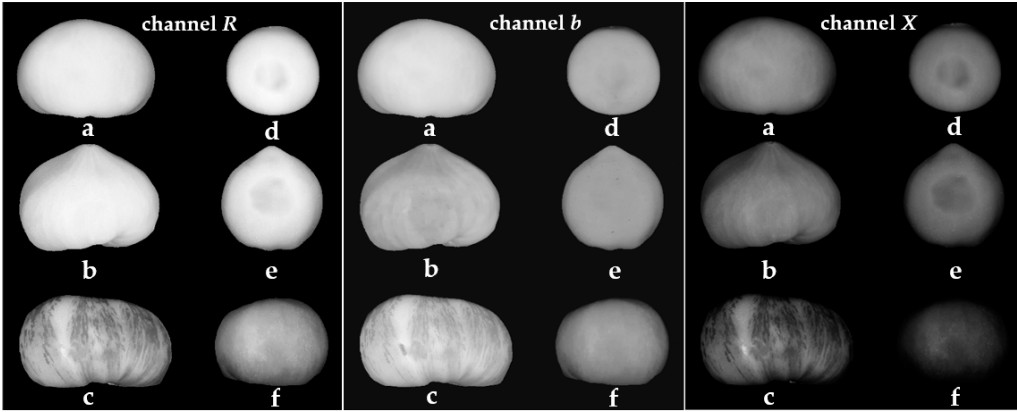

**Figure 2.** The images from selected color channels of tomato 'Ożarowski' (**a**), 'Marvel Striped' (**b**), 'Green Zebra' (**c**), Sandoline F1 (**d**), Cupidissimo F1 (**e**), and Sacher F1 (**f**).

*2.3. Lycopene Extraction*

Lycopene content was determined by the method reported by Bohoyo-Gil et al. [29]. Two grams of the ground sample was homogenized in the extraction solution (hexane:acetone 6:4) with the addition of 0.1 g of magnesium carbonate. The solution was filtered through a Büchner funnel under reduced pressure. The extract was transferred to a separating funnel, and 50 mL of water was added and shaken. After phase separation, the water–acetone phase was discarded. The acetone rinsing operation was repeated until the lower phase was free of acetone and the upper hexane phase containing lycopene was fil-

tered into an evaporation flask through a filter paper containing anhydrous sodium sulfate. Hexane was evaporated to dryness in a vacuum evaporator at 40 °C; the dry residue was quantitatively transferred to a 25 mL flask with a solution of acetonitrile:methanol:ethyl acetate 55:25:20 + 0.1% BHT + 1 mL TEA and 4 mL of hexane. The flask extract was filtered with a 45 μm PTFE filter into an amber bottle and analyzed by HPLC.

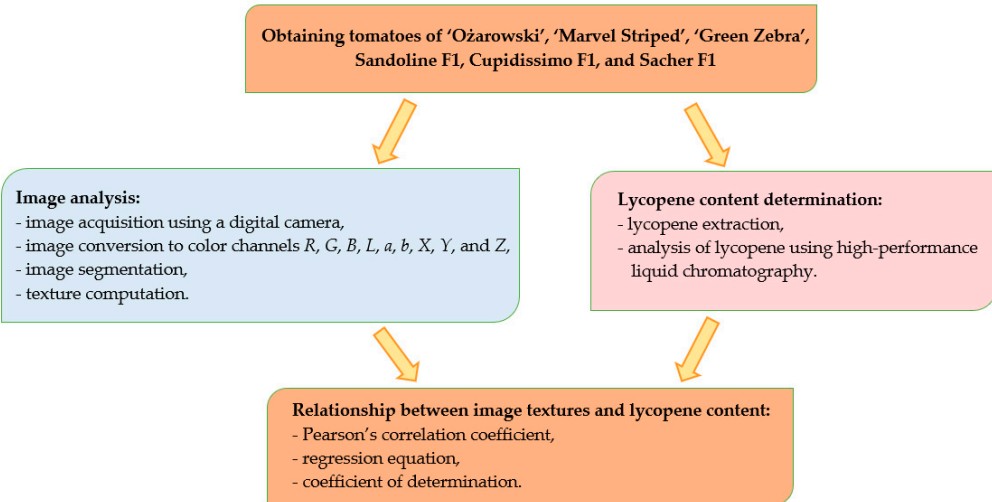

**Figure 3.** The flowchart presenting the stages for determining the relationship between image textures of tomato fruit and lycopene content.

### 2.4. HPLC Analysis of Lycopene

The content of lycopene in tomato samples was determined by high-performance liquid chromatography (HPLC). Separation was performed using a Kinetex C-18 column (250 mm × 4.6 mm; 5 μm) on an Agilent 1200 HPLC system equipped with a DAD detector. The elution conditions were as follows: 0.7 mL min$^{-1}$; temperature 28 °C; wavelength 472 nm; mobile phase: acetonitrile, ethyl acetate, methanol + 1 mL TEA + 1 g BHT in gradient flow. The calculations were made according to a standard curve for the lycopene standard (Sigma-Aldrich, Taufkirchen, Germany). Lycopene content was expressed in mg 100 g$^{-1}$.

### 2.5. Statistical Analysis

Statistical analysis was performed with STATISTICA 13.3 (StatSoft Polska Sp. z o.o., Kraków, Poland) as a one-way analysis of variance. The normality of the distribution was analyzed by Kolmogorov–Smirnov, Lilliefors, and Shapiro–Wilk tests. The homogeneity of variance was checked using Brown–Forsythe and Levene's tests. The significant differences between means were determined at $p = 0.05$ by Tukey's test. Lycopene content was conducted in triplicates ($n = 3$). The results are based on the fresh weight of the products obtained.

Additionally, the linear relationships between the lycopene content and the texture parameters of tomato fruit cultivars were determined using STATISTICA 13.3. Pearson's correlation coefficients (R) at a significance level of $p < 0.05$, regression equations, and coefficients of determination ($R^2$) for the lycopene content and the selected textural parameters were computed.

### 3. Results

The results of lycopene content for the six cultivars of tomato are shown in Table 1. The dependence of the lycopene content on the cultivar and thus on the color of the tomato fruit was observed. Four homogenous groups were found. Cultivars with red and brown fruit were characterized by the highest content of lycopene. Tomatoes belonging to Cupidissimo F1 (11.74 mg 100 g$^{-1}$) and Sacher F1 (11.83 mg 100 g$^{-1}$) were in one homogenous group.



The lowest lycopene content was found in the case of 'Green Zebra' with green fruit (0.31 mg 100 $g^{-1}$) and 'Ożarowski' with yellow fruit (0.37 mg 100 $g^{-1}$), which formed one homogenous group.

**Table 1.** The lycopene content in tomato fruit of various cultivars with different colors.

| Tomato Cultivar | Fruit Color | Lycopene Content (mg 100 $g^{-1}$) |
| --- | --- | --- |
| 'Ożarowski' | yellow | 0.37 d |
| 'Marvel Striped' | yellow-orange-pink | 1.19 c |
| 'Green Zebra' | green | 0.31 d |
| Sandoline $F_1$ | red | 9.89 b |
| Cupidissimo $F_1$ | red | 11.74 a |
| Sacher $F_1$ | brown | 11.83 a |

Note: Means in the column marked with the same letter are not different according to Tukey's HSD test ($p = 0.05$).

It was found that the difference in lycopene content was related to the values of the outer surface textures of tomato fruit images. Up to five textures from each color channel for which the highest statistically significant correlation coefficients greater than 0.80 were obtained without considering whether it was a positive or negative correlation were chosen to be presented in this paper. For some color channels, fewer than five textures were statistically significantly correlated with the lycopene content, or no texture was correlated with the lycopene content. In the case of RGB color space (Table 2), statistically significant correlation coefficients (R) were determined only for textures from color channels *G* and *B*. No texture from the color channel *R* was statistically significantly correlated with the lycopene content of tomato fruit. The highest values of the R parameter ($-0.99$) were obtained in the case of selected textures from color channel *G*. In the case of this color channel, all the highest correlation coefficients were negative. For color channel *B*, slightly lower negative values reaching $-0.97$ and positive values of correlation coefficient reaching 0.94 were determined. The scatter plots for lycopene content and one selected image texture of each color channel *G* and *B* of tomato fruit are presented in Figure 4. In the case of both textures, the linear negative relationships with lycopene content are visible.

**Table 2.** Correlation coefficients (R) between lycopene content (mg 100 $g^{-1}$) and selected image textures from RGB color space for 'Ożarowski', 'Marvel Striped', 'Green Zebra', Sandoline F1, Cupidissimo F1, and Sacher F1 tomatoes; $p < 0.05$.

| Texture Parameter | Correlation Coefficient |
| --- | --- |
| GHPerc90 | $-0.99$ |
| GHDomn01 | $-0.99$ |
| GHDomn10 | $-0.99$ |
| GS5SV1SumAverg | $-0.99$ |
| GS5SV3SumAverg | $-0.99$ |
| BHMaxm10 | 0.94 |
| BSGPerc50 | 0.94 |
| BSGPerc90 | 0.94 |
| BS4RZRLNonUni | $-0.96$ |
| BS4RNGLevNonU | $-0.97$ |

G—color channel *G*; B—color channel *B*; Perc—percentile; Domn—dominant; SumAverg—sum average; Maxm— maximum of moments; RLNonUni—run length nonuniformity; GLevNonU—gray-level nonuniformity.

Textures from Lab color space (Table 3) were statistically significant linearly negatively and linearly positively correlated with lycopene content, and the R parameter reached $-0.99$ (texture bHPerc99 from color channel *b*) and 0.92 (texture LHSkewness from color channel *L*), respectively. In the case of color channel *L*, the negative correlation reached $-0.97$. For color channel *a*, slightly lower correlation coefficients (R) than for color channels *L* and *b* were observed. A negative correlation reaching $-0.95$ (aS4RHShrtREmp) and positive correlation reaching 0.83 (aS5SV1InvDfMom) were determined. The scatter plots

shown in Figure 5 confirmed the highest correlation between lycopene content and texture from the color channel *b*.

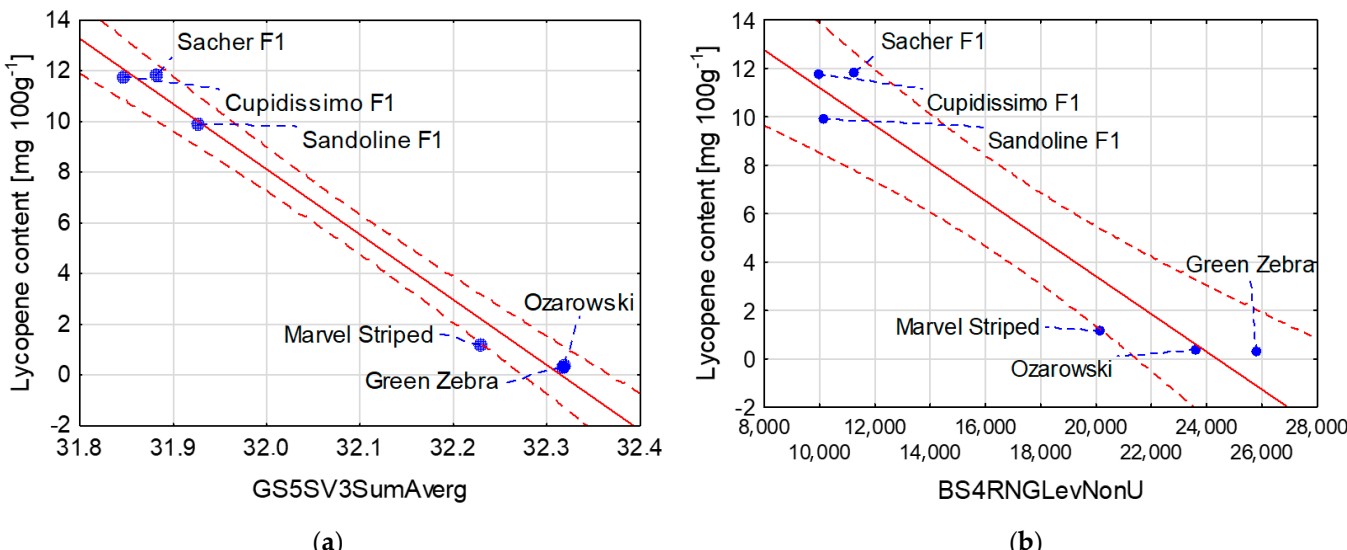

**Figure 4.** Scatter plots for lycopene content (mg 100 g$^{-1}$) and selected image textures: GS5SV3SumAverg (**a**) and BS4RNGLevNonU (**b**) from RGB color space of tomato fruit. Blue dot—mean value for the sample; blue dashed line—line connecting the mean value to the sample name; solid red line—regression line; red dashed line—confidence interval (95%).

**Table 3.** Correlation coefficients (R) between lycopene content (mg 100 g$^{-1}$) and selected image textures from Lab color space of 'Ożarowski', 'Marvel Striped', 'Green Zebra', Sandoline F1, Cupidissimo F1, and Sacher F1 tomatoes; $p < 0.05$.

| Texture Parameter | Correlation Coefficient |
| --- | --- |
| LHSkewness | 0.92 |
| LHPerc90 | −0.94 |
| LHPerc99 | −0.97 |
| LHDomn01 | −0.91 |
| LHDomn10 | −0.91 |
| aS5SV1InvDfMom | 0.83 |
| aS4RHShrtREmp | −0.95 |
| aS4RVShrtREmp | −0.94 |
| aS4RVFraction | −0.83 |
| aS4RNShrtREmp | −0.83 |
| bHPerc50 | −0.92 |
| bHPerc90 | −0.98 |
| bHPerc99 | −0.99 |
| bHDomn01 | −0.93 |
| bHDomn10 | −0.97 |

L—color channel *L*; a—color channel *a*; b—color channel *b*; Skewness—skewness coefficient; Perc—percentile; Domn—dominant; InvDfMom—inverse difference moment; ShrtREmp—short run emphasis; Fraction—fraction of image in runs.

The correlation coefficients between lycopene content and textures from XYZ color space (Table 4) were very high, reaching −0.99 for YHVariance from color channel *Y*. The highest positive correlation was 0.98 and also belonged to color channel *Y* (YHMaxm10). In the case of color channels *X* and *Z*, only two textures were statistically significantly correlated with lycopene content. The R parameter reached −0.98 for XHVariance and −0.93 for ZHDomn01. The highest correlations between lycopene content and textures from XYZ color space for textures from the color channels *X* and *Y* were confirmed by scatter plots, shown in Figure 6.

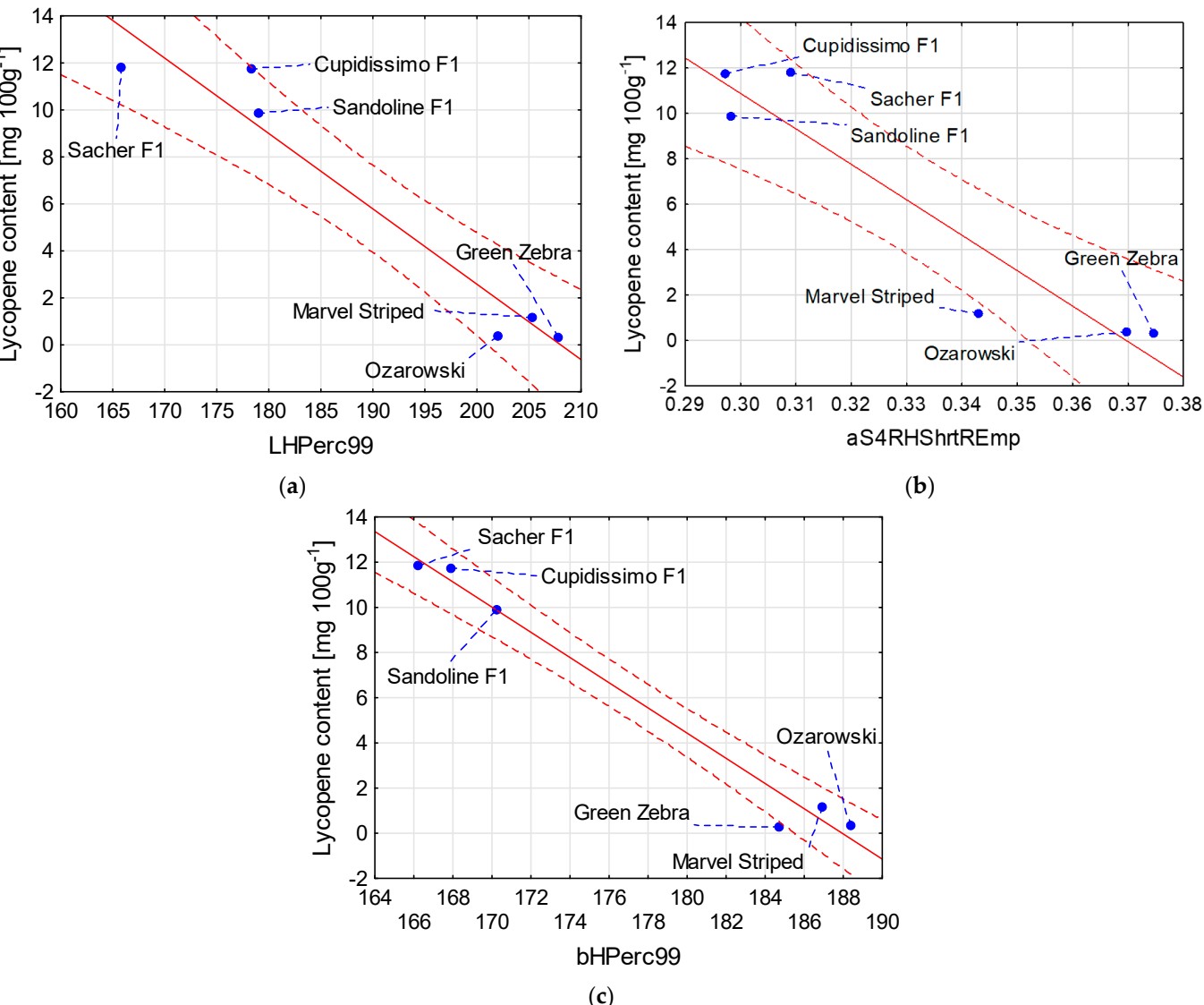

**Figure 5.** Scatter plots for lycopene content (mg 100 g$^{-1}$) and selected image textures: LHPerc99 (**a**), aS4RHShrtREmp (**b**) and bHPerc99 (**c**) from Lab color space of tomato fruit. Blue dot—mean value for the sample; blue dashed line—line connecting the mean value to the sample name; solid red line—regression line; red dashed line—confidence interval (95%).

**Table 4.** Correlation coefficients (R) between lycopene content (mg 100 g$^{-1}$) and selected image textures from XYZ color space of 'Ożarowski', 'Marvel Striped', 'Green Zebra', Sandoline F1, Cupidissimo F1, and Sacher F1 tomatoes; $p < 0.05$.

| Texture Parameter | Correlation Coefficient |
| --- | --- |
| XHVariance | −0.98 |
| XHMaxm10 | 0.87 |
| YHVariance | −0.99 |
| YHPerc90 | −0.97 |
| YHPerc99 | −0.97 |
| YHMaxm10 | 0.98 |
| YS5SN5SumOfSqs | −0.94 |
| ZHPerc50 | −0.90 |
| ZHDomn01 | −0.93 |

X—color channel *X*; Y—color channel *Y*; Z—color channel *Z*; Maxm—maximum of moments; Perc—percentile; SumOfSqs—sum of squares; Domn—dominant.

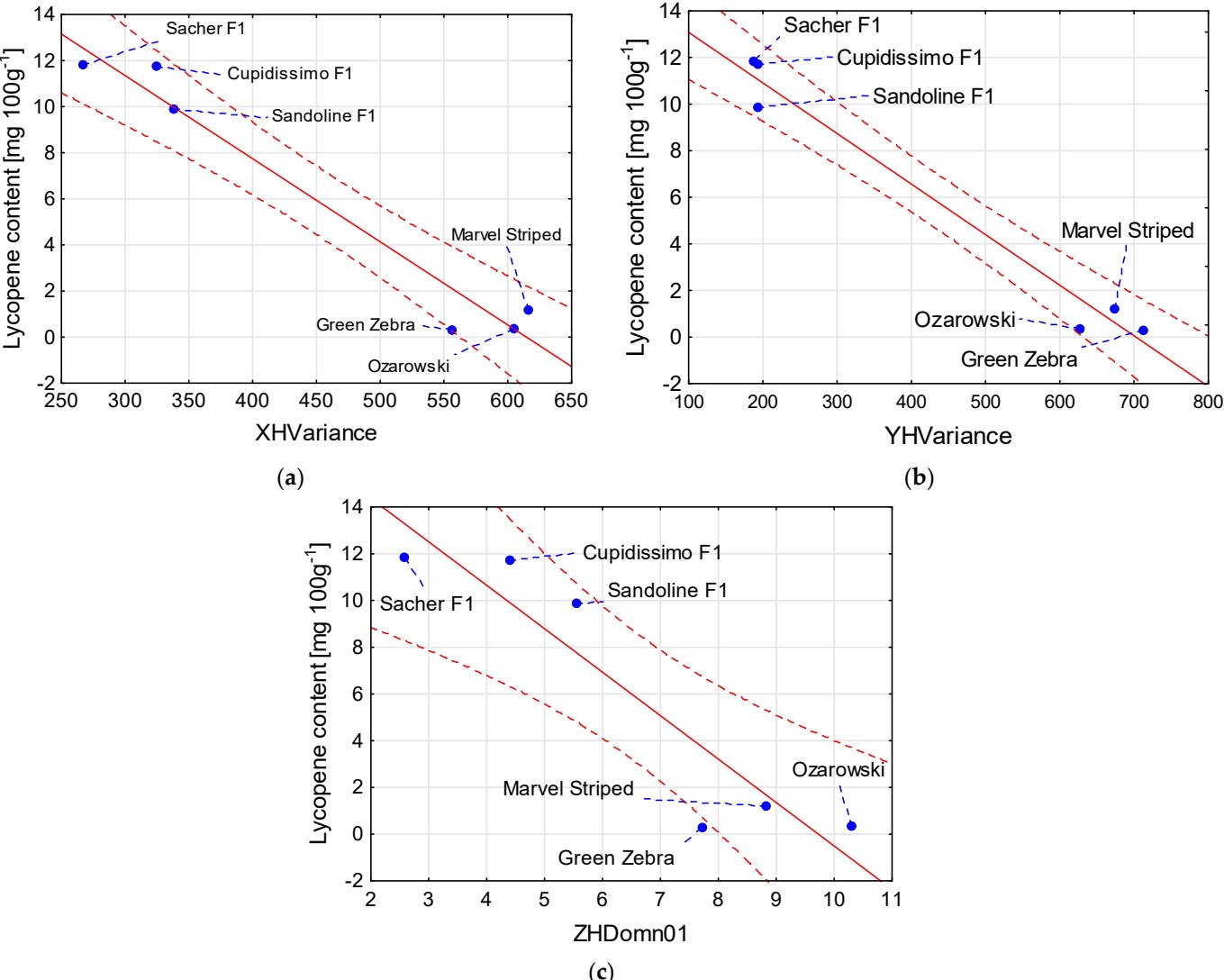

**Figure 6.** Scatter plots for lycopene content (mg 100 g$^{-1}$) and selected image textures: XHVariance (**a**), YHVariance (**b**) and ZHDomn01 (**c**) from XYZ color space of tomato fruit. Blue dot—mean value for the sample; blue dashed line—line connecting the mean value to the sample name; solid red line—regression line; red dashed line—confidence interval (95%).

Regression equations and coefficients of determination ($R^2$) for relationships between lycopene content and selected texture parameters are presented in Table 5. In the case of individual color channels, one texture with the highest correlation coefficient (Tables 2–4) and providing the highest results of regression equations and coefficients of determination was selected. The $R^2$ value of 0.99 was reached in the case of texture from color channel *G* (GS5SV3SumAverg). Additionally, a very high value of the coefficient of determination ($R^2$) equal to 0.98 was obtained for textures from color channels *b* (bHPerc99) and *Y* (YHVariance), 0.96—for texture from color channel *X* (XHVariance), and 0.94—for textures from color channels *B* (BS4RNGLevNonU) and *L* (LHPerc99). These indicate a high accuracy for the derived regression equation.

**Table 5.** The regression equations between lycopene content and selected image textures of 'Ożarowski', 'Marvel Striped', 'Green Zebra', Sandoline F1, Cupidissimo F1, and Sacher F1 tomatoes.

| Regression Equation | Coefficient of Determination ($R^2$) |
|---|---|
| Lycopene content [mg 100 g$^{-1}$] = 831.95 – 25.74 × GS5SV3SumAverg | 0.99 |
| Lycopene content [mg 100 g$^{-1}$] = 18.994 – 0.0008 × BS4RNGLevNonU | 0.94 |
| Lycopene content [mg 100 g$^{-1}$] = 66.744 – 0.3208 × LHPerc99 | 0.94 |
| Lycopene content [mg 100 g$^{-1}$] = 57.754 – 156.2 × aS4RHShrtREmp | 0.90 |
| Lycopene content [mg 100 g$^{-1}$] = 104.82 – 0.5577 × bHPerc99 | 0.98 |
| Lycopene content [mg 100 g$^{-1}$] = 22.166 – 0.0361 × XHVariance | 0.96 |
| Lycopene content [mg 100 g$^{-1}$] = 15.251 – 0.0217 × YHVariance | 0.98 |
| Lycopene content [mg 100 g$^{-1}$] = 18.090 – 1.860 × ZHDomn01 | 0.86 |

G—color channel *G*; B—color channel *B*; L—color channel *L*; a—color channel *a*; b—color channel *b*; X—color channel *X*; Y—color channel *Y*; and Z—color channel *Z*; SumAverg—sum average; GLevNonU—gray level nonuniformity; Perc—percentile; ShrtREmp—short run emphasis; Domn—dominant.

## 4. Discussion

In the research performed, the tested tomato cultivars differed significantly in the content of lycopene. The content of lycopene in the tested samples ranged from 0.31 in the fruit of 'Green Zebra' tomato to 11.83 mg 100 g$^{-1}$ in the fruit of the Sacher F$_1$ cultivars. Color is generally an accurate indicator of the lycopene content. The results of other authors confirm that the content of lycopene in tomato fruits depends on the cultivar and may be within a wide range [30–34]. The darker the red color, the more lycopene in tomato fruits, and the less β-carotene. Brown fruits of tomato fruits have the most lycopene and little β-carotene [20]. Additionally, our research found that red and brown cultivars contained more lycopene than cultivars with other colors. The content of this compound in tomato fruit depends not only on the varietal characteristics that determine the color of ripe fruit, but also on the place of cultivation, fertilization, and agroclimatic conditions. The temperature during the growth of tomato vegetation is of significant importance, and for the optimal content of lycopene, during fruit growth, it should be in the range 16–22 °C. Significant increase in temperature, above 35 °C, causes the conversion of lycopene into β-carotene [35]. The content of lycopene in plant tissues depends on many factors and may undergo changes not only in living plants, but also during their processing and storage [35–39].

The results indicated the usefulness of texture parameters obtained using image processing for the estimation of the lycopene content in fruit. The previous studies reported in the literature also provided a statistically significant correlation between image textures and other properties of plant material. For example, image textures of wheat kernels were highly correlated with the quantity of DNA of fungi of the genus *Fusarium*. For the ventral side of kernels, the correlation coefficient (R) reached 0.86 in the case of texture from channel *a* in cultivar 1 and 0.80 for texture from channel *R* in cultivar 2. In the case of images of the dorsal side of wheat kernels, the highest correlation coefficient of 0.89 was obtained for texture from channel *V* in cultivar 3 [40]. Nazari et al. [41] correlated image textures and the content of phenolic compounds, tannin, and protein of sorghum grain. The protein content was most highly correlated with texture from channel *L* and the correlation coefficient was equal to 0.83. The highest correlation coefficients for tannin content of −0.91 and total phenolic content of −0.94 were determined for textures from color channel *S* [41]. The satisfactory results of the present study and the abovementioned literature data on the existence of correlations between the image textures and other features, including the chemical properties of plant material, suggest further research in this field. First, more cultivars of tomato and cherry tomato can be included. Additionally, research using other species of fruit and vegetables can be carried out.

## 5. Conclusions

This study revealed the relationship between image textures and lycopene content, determined in a nondestructive, objective, and simple way. This contrasts with lycopene content determined by high-performance liquid chromatography, which is destructive, more expensive, and time-consuming. High correlation coefficients (R) between lycopene content and texture parameters reaching $-0.99$ were obtained. It was found that textures from color channel *G* were the most useful for the estimation of lycopene content. The models predicted the lycopene content in tomato fruit with a coefficient of determination ($R^2$) reaching 0.99. It was concluded that the lycopene content may be estimated with high precision using determined regression equations based on image textures. The results can have practical application in tomato processing and consumption for the selection of fruit with desirable lycopene content. Further research may include more cultivars of tomato and cherry tomato.

**Author Contributions:** Conceptualization, E.R. and J.S.-G.; methodology, E.R. and J.S.-G.; software, E.R.; validation, E.R.; formal analysis, E.R. and J.S.-G.; investigation, E.R. and J.S.-G.; resources, E.R. and J.S.-G.; data curation, E.R. and J.S.-G.; writing—original draft preparation, E.R. and J.S.-G.; writing—review and editing, E.R.; visualization, E.R.; supervision, E.R. All authors have read and agreed to the published version of the manuscript.

**Funding:** This work was performed in the frame of multiannual programme "Actions to improve the competitiveness and innovation in the horticultural sector with regard to quality and food safety and environmental protection", task 3.5. Development of innovative technologies for the storage and use of fruit and vegetables (task leader: Krzysztof Rutkowski, PhD) financed by the Polish Ministry of Agriculture and Rural Development.

**Institutional Review Board Statement:** Not applicable.

**Data Availability Statement:** The data presented in this study are available on request from the corresponding author.

**Conflicts of Interest:** The authors declare no conflict of interest.

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
