# Peer review of "Relationship of Textures from Tomato Fruit Images Acquired Using a Digital Camera and Lycopene Content Determined by High-Performance Liquid Chromatography"

_agriculture, doi:10.3390/agriculture12091495_

Round 1

Reviewer 1 Report

Overall the article is easy to read, and shows interesting correlation between image texture and the lycopene content of tomato fruits. However, few things could be improved:

The correlation is made between the "image texture" and chemical substance, but there is no explanation in the Introduction or the Materials and Methods on the definition of image texture. It would be helpful to provide an explanation on the meaning of this for non-image processing background readers.

Under section 2.2 Image Analysis it would be more interesting to share example photos of the tomatoes taken in the black box with LED illumination as mentioned. The whole paper has not a single photo of a tomato.

The results presented in Table 2 is a bit confusing with undefined texture parameters. For example, GHPerc90's only mention is in Table 2 and is not mentioned anywhere else in the whole article. Are these standard notations? A bit of an explanation of what GHPerc90, GHDomn01, etc would be helpful.

I have the same issues with the results presented in Table 3 and 4.  The texture parameters need explanation as names like YS5SN5SumOfSqs really confusing the readers.

Author Response

- Overall the article is easy to read, and shows interesting correlation between image texture and the lycopene content of tomato fruits. However, few things could be improved:

Answer: Thank you very much for your careful reading of our manuscript. All your comments have been considered and the manuscript has been improved.

- The correlation is made between the "image texture" and chemical substance, but there is no explanation in the Introduction or the Materials and Methods on the definition of image texture. It would be helpful to provide an explanation on the meaning of this for non-image processing background readers.

Answer: We apologize for this mistake. The definition of image texture has been included in the Materials and Methods as follows:

" Textures determined in this study are considered as numerical data extracted from images and are defined as a function of the spatial variation of the brightness intensity of the pixels. Objects can be characterized by different textures even if they have the same color histograms and number of pixels but a dissimilar color distribution. These changes can be difficult to perceive visually [27,28]." (lines 152-157)

- Under section 2.2 Image Analysis it would be more interesting to share example photos of the tomatoes taken in the black box with LED illumination as mentioned. The whole paper has not a single photo of a tomato.

Answer: This is a very useful comment. The original images have been added in Figure 1 (Figure 1 The sample color images of tomatoes ‘Ożarowski’ (a), ‘Marvel Striped’ (b), ‘Green Zebra’ (c), Sandoline F1 (d), Cupidissimo F1 (e), and Sacher F1 (f).)

- The results presented in Table 2 is a bit confusing with undefined texture parameters. For example, GHPerc90's only mention is in Table 2 and is not mentioned anywhere else in the whole article. Are these standard notations? A bit of an explanation of what GHPerc90, GHDomn01, etc would be helpful.

Answer: These names are specific to parameters calculated using MaZda. The texture names have been explained below each table.

- I have the same issues with the results presented in Table 3 and 4.  The texture parameters need explanation as names like YS5SN5SumOfSqs really confusing the readers.

Answer: As above - The texture names have been explained below each table.

Reviewer 2 Report

The submitted manuscript is well written and provide interesting information about the correlations between image textures and lycopene content of tomato fruit belonging to different cultivars.

My main concerns are about the way the Authors obtained the tomato samples, and the data presentation.

1.     The Authors mention that the fruits were purchased from the producer. They should add info about the growing period. Not knowing the growing history of your materials can have important impacts on the lycopene content.

 2.     The authors mention that they used a digital camera to take the images. How reproducible are the data or images taken with the digital camera? Was any type of adjustment or calibration made to standardize the taking of the images?

 3.     In the results section, the authors reported the correlation coefficients between lycopene content and textures in different color spaces (RGB, Lab, XYZ). Most of the correlation coefficient values shown in the tables gave negative values, and only few correlation coefficient values were positive. On what were the authors based to select the texture parameters shown in table 5? What are the characteristics of these texture parameters that present high values of the determination coefficient in relation to lycopene content? ¿Can this type of program based on image textures differentiate between isomers of a compound? Please explain

Author Response

Reviewer 2

The submitted manuscript is well written and provide interesting information about the correlations between image textures and lycopene content of tomato fruit belonging to different cultivars.

Answer: We are grateful to the Reviewer for this comment.

My main concerns are about the way the Authors obtained the tomato samples, and the data presentation.

Answer: Thank you very much for this comment. We totally agree with this opinion. Sections 2. Materials and Methods and 3. Results have been expanded as indicated below.

  1. The Authors mention that the fruits were purchased from the producer. They should add info about the growing period. Not knowing the growing history of your materials can have important impacts on the lycopene content.

Answer: More detailed information has been added as follows:

“The tomatoes from potted seedlings were grown in an unheated tunnel from April to September. All tomato fruits cultivar were purchased from the producer in the last week of June, in the consumption maturity phase.” (lines 116-119)

  1. The authors mention that they used a digital camera to take the images. How reproducible are the data or images taken with the digital camera? Was any type of adjustment or calibration made to standardize the taking of the images?

Answer: Image acquisition has been described in more detail as follows:

“The digital camera included Optical Image Stabilization, Auto White Balance, F 2.4, 8x digital zoom, and SD (Secure Digital) card. The LED illumination was characterized by the light sources of 24 LED, Related Input Voltage of AC110-240 V/50–60 Hz, Related In-put current of 0.07 A, and Related Output Power of 2.2 W [25].” (lines 123-127)

  1. In the results section, the authors reported the correlation coefficients between lycopene content and textures in different color spaces (RGB, Lab, XYZ). Most of the correlation coefficient values shown in the tables gave negative values, and only few correlation coefficient values were positive. On what were the authors based to select the texture parameters shown in table 5? What are the characteristics of these texture parameters that present high values of the determination coefficient in relation to lycopene content? ¿Can this type of program based on image textures differentiate between isomers of a compound? Please explain

Answer: It has been explained in detail as follows:

“Up to 5 textures from each color channel for which the highest statistically significant correlation coefficients greater than 0.80 were obtained without considering whether it was a positive or negative correlation were chosen to be presented in this paper. For some color channels, fewer than five textures were statistically significantly correlated with the lycopene content, or no texture was correlated with the lycopene content.” (lines 212-217)

“Regression equations and coefficients of determination (R2) for relationships between lycopene content and selected texture parameters are presented in Table 5. In the case of individual color channels, one texture with the highest correlation coefficient (Tables 2-4) and providing the highest results of regression equations and coefficients of determination was selected.” (lines 284-288)

During the analysis, the total lycopene content was considered without considering the isomers.

Reviewer 3 Report

Summary:

The manuscript used high-performance liquid chromatography (HPLC) to determine the lycopene content, and the image features of tomato fruits were associated with the lycopene content. Specifically, tomato fruit varieties of various colors were selected for the work. The overall work was excellent, and the results were encouraging. However, some issues are of concern.

Issues:

1. The introduction of a lack of HPLC in the quantitative analysis of fruit application, especially lycopene content reference.

2. The image with intuitive show a lack of material, especially tomato fruits images of the different color channel.

3. The authors are suggested to provide a flowchart of the steps involved in the proposed method for better understanding.

4. Color is related to lycopene content. The regression equation based on feature decoupling to predict lycopene content further enhances the scientific and interesting nature of the manuscript and supports the correlation between color features and lycopene. In other words, regression equations modeled on data from red fruits predicted lycopene content in green fruits and vice versa.

Author Response

Summary:

The manuscript used high-performance liquid chromatography (HPLC) to determine the lycopene content, and the image features of tomato fruits were associated with the lycopene content. Specifically, tomato fruit varieties of various colors were selected for the work. The overall work was excellent, and the results were encouraging. However, some issues are of concern.

Answer: Thank you for your careful reading of the manuscript and your valuable comments. The revised version of the manuscript has been improved.

Issues:

  1. The introduction of a lack of HPLC in the quantitative analysis of fruit application, especially lycopene content reference.

Answer: It has been added as follows: “The carotenoid content can be determined, for example, using spectrophotometric measurements, but high-performance liquid chromatography (HPLC) can be characterized by the higher accuracy for identifying and quantifying individual carotenoids from the extracts [11]. The lycopene content in tomatoes is commonly determined by HPLC [12,13].” (lines 53-57)

  1. Szabo, K.; Teleky, B.-E.; Ranga, F.; Roman, I.; Khaoula, H.; Boudaya, E.; Ltaief, A.B.; Aouani, W.; Thiamrat, M.; Vodnar, D.C. Carotenoid Recovery from Tomato Processing By-Products through Green Chemistry. Molecules 2022, 27, 3771.
  2. Kyriakoudi, A.; Tsiouras, A.; Mourtzinos, I. Extraction of Lycopene from Tomato Using Hydrophobic Natural Deep Eutectic Solvents Based on Terpenes and Fatty Acids. Foods 2022, 11, 2645.
  3. Mare, R.; Maurotti, S.; Ferro, Y.; Galluccio, A.; Arturi, F.; Romeo, S.; Procopio, A.; Musolino, V.; Mollace, V.; Montalcini, T.; Pujia, A. A Rapid and Cheap Method for Extracting and Quantifying Lycopene Content in Tomato Sauces: Effects of Lyco-pene Micellar Delivery on Human Osteoblast-Like Cells. Nutrients 2022, 14, 717.

  1. The image with intuitive show a lack of material, especially tomato fruits images of the different color channel.

Answer: The original (color) tomato images have been added in Figure 1 (Figure 1 The sample color images of tomatoes ‘Ożarowski’ (a), ‘Marvel Striped’ (b), ‘Green Zebra’ (c), Sandoline F1 (d), Cupidissimo F1 (e), and Sacher F1 (f)) and the images from selected color channels have been added in Figure 2 (Figure 2 The images from selected color channels of tomato ‘Ożarowski’ (a), ‘Marvel Striped’ (b), ‘Green Zebra’ (c), Sandoline F1 (d), Cupidissimo F1 (e), and Sacher F1 (f)).

  1. The authors are suggested to provide a flowchart of the steps involved in the proposed method for better understanding.

Answer: The flowchart has been provided in Figure 3 (Figure 3 The flowchart presenting the stages of determining the relationship between image textures of tomato fruit and lycopene content).

  1. Color is related to lycopene content. The regression equation based on feature decoupling to predict lycopene content further enhances the scientific and interesting nature of the manuscript and supports the correlation between color features and lycopene. In other words, regression equations modeled on data from red fruits predicted lycopene content in green fruits and vice versa.

Answer: Thank you very much for this opinion. This was exactly the intention of our study.

Round 2

Reviewer 2 Report

The Authors addressed all my comments and the paper is now improved and can be accepted in the present form.